# The Effect of Black Garlic on the Volatile Compounds in Heat-Treated Sucuk

**DOI:** 10.3390/foods12203876

**Published:** 2023-10-23

**Authors:** Zeynep Feyza Yılmaz Oral, Güzin Kaban

**Affiliations:** 1Department of Food Technology, Vocational College of Technical Sciences, Atatürk University, Erzurum 25240, Türkiye; 2Department of Food Engineering, Faculty of Agriculture, Atatürk University, Erzurum 25240, Türkiye; gkaban@atauni.edu.tr

**Keywords:** garlic, black garlic, volatile compounds, heat-treated sucuk, fermented sausage

## Abstract

This study aimed to determine the influence of using black garlic (BG) at different levels on organic volatile compounds in heat-treated sucuk (HTS), a semi dry fermented sausage. Three independent batches of sausages were prepared: control: 1% white garlic (WG): WG-1%; BG-1%: 1% BG; BG-2%: 2% BG; and BG-3%: 3% BG. After stuffing, the sausages were subjected to fermentation, heat treatment (internal temperature of 64 °C), and drying, respectively. After production, the final products were analyzed for volatile compounds. A solid-phase microextraction technique was used for the extraction of volatile compounds, and identification was carried out by a gas chromatograph/mass spectrometer. A total of 47 volatile compounds, including sulfur compounds, alcohols, esters, ketones, aliphatic hydrocarbons, acids, aromatic hydrocarbons, aldehydes, and terpenes, were identified from the sausages. The use of BG in HTS had no significant effect on aliphatic hydrocarbons, acids, ketones, aromatic hydrocarbons, and esters. BG increased the abundances of 2-propen-1-ol, allyl methyl sulfide, methyl 2-propenyl-disulfide, sabinene, β-pinene, and β-phellandrene regardless of the increase in the addition rate. BG-3% increased the level of hexanal. According to the PCA results, BG-containing groups showed positive correlation with esters, sulfur compounds, aldehydes, aromatic hydrocarbons, and alcohols, but these chemical groups were more closely correlated with BG-3%. In addition, diallyl disulfide, which is one of the main factors that causes the pungent and spicy smell of garlic, showed a close correlation with WG-1%.

## 1. Introduction

Fermented sausages have a special importance for consumers not only for their typical flavor but also for the fact that they are considered safe and healthy foods. Yet, strategies are being developed to reduce synthetic additives or to use natural products with functional components in order to increase the value-added nature of these products [1]. In this context, black garlic is among the most remarkable natural products. Black garlic, also known as fermented garlic, is a new garlic product that has become popular in recent years and is produced by aging white garlic at high temperatures and relative humidity for 3–4 months [2].

The main reaction during the processing of black garlic is the Maillard reaction, which causes the color of the garlic to change from white to black. Some compounds that are not originally present in white garlic are also formed during the process [3,4,5], and polysaccharide degradation occurs during heat treatment [4]. At the same time, the pungent smell of white garlic is minimized as a result of fermentation, and a chewy and jellylike texture is obtained [3,4,5].

Allicin is responsible for the intense smell and spiciness of freshly crushed garlic. Allicin, one of the organosulfur compounds catalyzed by alliinase [6], the major thiosulfinate in fresh garlic, is very unstable and converts to a wide variety of sulfur derivatives, most of them volatiles [5]. However, the pungent taste caused by allicin can significantly limit the use and, therefore, consumption of fresh garlic in different areas, including gastronomy, and this situation increases the demand for alternative garlic products, such as black garlic, which do not have an intense odor [6].

White garlic is a spice that finds its way into the formulation of many fermented sausages. In Türkiye, fermented sausages called sucuk and heat-treated sucuk (HTS) are preferred by most consumers, and garlic is generally used at around 1% in the formulation of both types of fermented sausage. Sucuk is a type of dry-fermented sausage, and its production is based on fermentation and drying and does not include smoke or heat treatment [7]. The HTS is a type of semidry fermented sausage that is produced on an industrial scale from beef and beef fat through short-term fermentation, heat treatment (internal temperature 60–68 °C), and drying. The pH value for this product is a maximum of 5.6 [8].

Garlic is used at even higher levels in traditional sucuk formulations in different localities. Garlic is very effective in the formation of the characteristic flavor of both sucuk and HTS, and it is the main source of sulfur compounds [8,9]. It is reported that the concentration of sulfur compounds decreases, but the levels of roasted and sweet volatiles increase during the fermentation of black garlic [5]. However, it is also stated that different volatile compounds are formed in black garlic at different concentrations depending on the processing conditions [5,10,11]. For this reason, it is of great importance to determine the effect of black garlic on the volatile profile in the fermentation environment. Although there is a study on the effect of BG on nitrosamine formation and some product properties in HTS [12], there is no study on the effect of black garlic on the volatile compounds in fermented sausages including HTS. In the current study, we aimed to determine the effects of using different amounts of black garlic on the volatile compound profile of HTS.

## 2. Materials and Methods

### 2.1. Material

Fresh lean beef and beef fat were purchased from a local butcher (Erzurum, Türkiye). White garlic (WG) and black garlic (BG) were purchased from a commercial company (Taşköprü, Kastamonu, Türkiye) (Figure 1). Autochthonous *Latiplantibacillus plantarum* GM77 and *Staphylococcus xylosus* GM92 [13] were added to batters as starter cultures at levels of 10^7^ and 10^6^ cfu/g, respectively.

### 2.2. Preparation of Heat-Treated Sucuk (HTS)

HTS manufacture was carried out in the meat-processing unit of the Department of Food Engineering, Atatürk University (Erzurum, Türkiye). Three independent of batches of sausages were prepared, and a total four groups of sausages were manufactured in each batch: 1% white garlic (WG-1%) (control); 1% black garlic (BG) (BG-1%); 2% BG (BG-2%); and 3% BG (BG-3%). Thus, a total of 12 batters were prepared in three productions.

The sausage batters were prepared with lean beef (80%) (3.2 kg) and beef fat (20%) (0.8 kg) using a cutter (Mado MTK 662, Dornhan, Schwarzwald, Germany). The following ingredients were used per kg of meat and fat: 20 g NaCl, 7 g red pepper, 2.5 g allspice, 5 g black pepper, 9 g cumin, 4 g sucrose, and 0.15 g sodium nitrite. Starter cultures were also added during the preparation of the batters. For the control treatment, 1% white garlic was added to the batters. For other treatments, 1%, 2%, or 3% black garlic was used. Collagen casings (38 mm ø, Naturin GmbH, Weinheim Germany) were filled with the prepared batters with a filling machine (Mado MTK 591, Dornhan, Schwarzwald, Germany). Subsequently, the HTS samples were fermented in a climate unit (Reich, Thermoprozestechnik GmbH, Schechingen, Germany) that maintained a temperature of 24 ± 1 °C and a relative humidity of 90 ± 2%. The fermentation was continued until the pH reached 5.3 (about 20 to 22 h); pH was analyzed with a pH meter (205, Testo, Lenzkirch, Germany). After this stage, HTS groups were cooked in a steam-cooking unit (Mauting, Valtice, Czechia) to a core temperature of up to 64 °C. Finally, the groups were dried in the climate unit (16 ± 1 °C, 84% ± 2 relative humidity) until water activity reached 0.93. The water activity was measured with a water activity device (TH-500, Novasina, Lachen, Switzerland).

### 2.3. Volatile Compounds Analysis

For extraction, a 40 mL vial containing 5 g of sample was kept in a thermal block at 30 °C for 1 h, allowing headspace collection of volatile compounds. A solid-phase microextraction method was used to extract the volatile compounds, and 75 μm of Carboxen/polydimethylsiloxane (CAR/PDMS fiber, Supelco, Bellefonte, PA, USA) was placed in vials and kept for 2 h. The volatile compounds adsorbed by the fiber were desorbed using the injection port of a gas chromatograph/mass spectrometer detector (GC 6890N/MS 5973, Agilent, Santa Clara, CA, USA) for 6 min at 250 °C using the splitless mode.

The separation was performed on a DB-624 column (J&W Scientific, Folsom, CA, USA, 30 m × 0.25 mm × 1.4 μm), and helium was used as the carrier gas at a flow rate of 1 mL/min. After the fiber was inserted, the GC oven temperature program was started and ran as follows: at 40 °C for 5 min, 40 to 110 °C at 3 °C /min, 110 °C to 150 °C at 4 °C/min, then 150 °C to 210 °C at 10 °C /min; it was then kept at this temperature for 12 min. The interface of the GC/MS was maintained at 280 °C. The mass spectra were obtained in the electron impact mode at 70 eV. The data were acquired across the range of 30 to 400 amu.

The identification of volatile compounds was achieved by comparing mass spectral data obtained from NIST, WILEY, and FLAVOR libraries, by comparing the retention times and mass spectra of volatiles to those of authentic compounds, and by determining Kovats indexes (Parrafine mix, Supelco 44585-U, Bellefonte, PA, USA) and comparing them with those reported in the literature. All measurements were performed in triplicate. The results were given in AU × 10^6^ [8].

### 2.4. Statistical Analysis

Three independent batches were prepared on different days (replications). A total of four groups of sausages were manufactured in each batch, and the experiments were carried out according to the randomized complete block design. Data were evaluated by ANOVA using a general linear model, with the use of black garlic as a main effect and replication as a random effect. Duncan’s multiple range test was used to determine differences between means (*p* < 0.05 level). Statistical analyses were performed using SPSS version 20 (Chicago, IL, USA). To determine the relationship between black garlic and volatile compounds, principal component analysis was also performed using Minitab 17.1.0 software (State College, PA, USA).

## 3. Results and Discussion

### 3.1. Volatile Compounds

A total of 47 volatile compounds were identified in the HTS groups, including five alcohols, three aliphatic hydrocarbons, five sulfur compounds, one acid, six aldehydes, three ketones, four aromatic hydrocarbons, 18 terpenes, and two esters (Table 1).

Alcohols are formed through pathways such as the metabolism of amino acids, lipid oxidation, and the reduction in methyl ketone [14,15]. Five alcohols were detected in the present study (Table 1). Among the detected alcohols, only 2-propen-1-ol was affected by BG treatment. The lowest level of this alcohol was observed in the WG-1% group. However, no statistically significant differences were determined between groups containing black garlic (Table 1). It was reported that 2-propen-1-ol, also known as allyl alcohol, is a compound to be considered as the flavor component of black garlic, and that this compound is formed from alliin during the processing of garlic at high temperatures [16]. Martinez-Casas et al. [17] found that this alcohol is the most abundant volatile compound in black garlic. It was also reported by Martinez-Casas et al. [17] that very high amounts of allyl alcohol are formed when garlic or alliin is heated at cooking temperatures. In other words, this alcohol is formed from alliin during the heat treatment of white garlic. In the current study, it is thought that the high level of this compound in the groups containing black garlic was due to the high temperature applied in the production of black garlic. In addition, the presence of this compound, which was determined at a certain level in the control group, was affected by the heat treatment applied in HTS production. The lowest mean level in terms of total alcohol abundance was determined in the WG-1% group, but the average value was not different from the BG-1% group. On the other hand, although the alcohol abundance increased as the BG usage rate increased, there was no statistical difference between the BG-1%, BG-2%, and BG-3% groups (Figure 2). In previous studies on HTS, among alcohols, ethanol was found to be the most abundant compound [8,18,19,20], which is a product of carbohydrate metabolism [21]. Furthermore, 1-propen-2-ol was identified in HTS by Sallan et al. [8] and Öz et al. [20].

Three aliphatic hydrocarbons (hexane, undecane, and dodecane) were determined in the HTS. The use of BG had no effect on these compounds (Table 1). At the same time, no significant differences were observed between treatments in the total abundance of aliphatic hydrocarbons (Figure 1). The influence of aliphatic hydrocarbons on the aroma profile is quite low due to their high threshold values [19]. Some aliphatic hydrocarbons were also reported in other studies on HTS [8,18,20].

In the current study, five sulfur compounds were detected (Table 1). They can originate from garlic and are also formed as a result of amino acid catabolism [8]. Due to their low threshold values, these compounds may affect the sensory characteristics of the meat product [22]. The use of BG had a very significant effect (*p* < 0.01) on allyl mercaptan (2-propene-1-thiol). The BG-2% and BG-3% groups gave a lower abundance than the WG-1% and BG-1% groups. This result is due to the fact that black garlic contains little or no allyl mercaptan. In a study, Martinez-Casas et al. [17] reported that this compound was not detected in black garlic. Similarly, in another study, allyl mercaptan was not detected in black garlic [23]. In addition, Kim et al. [24] reported that allyl mercaptan was more strongly correlated with white garlic. Allyl mercaptan was also identified in HTS by Sallan et al. [8], Armutçu et al. [18], and Yılmaz Oral and Kaban [19].

The BG increased the amount of allyl methyl sulfide (*p* <0.05). However, no significant differences were observed between BG treatments (*p* >0.05). Previous studies also showed that HTS contains allyl methyl sulfide due to the use of garlic [8,18,19,20]. Black garlic production conditions affect this compound. Setiyoningrum et al. [23] reported that the amount of allyl methyl sulfide increased after 21 days of heat treatment. Likewise, black garlic was reported to contain more sulfur than white garlic [5]. Kim et al. [24] investigated the influence of garlic on volatile compounds and reported that aged-crushed black garlic contained more allyl methyl sulfide than a raw garlic clove and raw-crushed garlic. Another sulfur compound commonly found in HTS is 3,3′-thiobis-1-propene [8,19,20]. As seen in Table 1, the use of black garlic in HTS production had no significant effect on this compound (*p* > 0.05). Methyl 2-propenyl-disulfide was very significantly (*p* < 0.05) affected by the use of BG. However, increasing the level of BG did not result in an increase in the level of this compound (*p* > 0.05). Like 3,3′-thiobis-1-propene, this compound is also commonly found in HTS [18,19,20]. In a study by Setiyoningrum et al. [23], methyl-2-propenyl disulfide was not detected in fresh garlic but was determined at the end of heat treatment. In the present study, the use of black garlic resulted in a decrease in diallyl disulfide levels (Table 1). Diallyl disulfide is a major sulfur compound in garlic and has many biological functions [25]. It has also been shown in other studies that this compound comprises a significant proportion of the sulfur compounds found in the HTS [8,18]. However, it was stated that the level of this compound is high in white garlic and decreases in black garlic [5,24]. On the other hand, it was also noted that although white garlic contains a high concentration of sulfur volatiles, the concentration of sulfur volatiles is reduced in black garlic due to fermentation [26].

Acetic acid can be formed by microorganisms in fermented meat products as well as by fatty acid oxidation, carbohydrate fermentation, or alanine catabolism [27]. On the other hand, it was stated that organic acids such as acetic acid detected in black garlic originate from the Maillard reaction or lipid oxidation [11]. In this study, acetic acid was detected, but there was no statistical difference between the treatments (Table 1). Acetic acid was also found in studies conducted on both sucuk [28] and HTS [18,19]. It was also reported that autochthonous starter cultures (*L. plantarum* and *S. xylosus*) increased acetic acid content in HTS [8].

Six aldehydes (3-methyl-butanal, pentanal, decanal, hexanal, nonanal, and 2-methyl-3-phenyl-propanal) were identified. The BG had a significant effect on pentanal. However, only the BG-3% group gave a higher value than the control. The lowest value for decanal was determined in the control group, and differences between other groups were not found to be significant. The highest hexanal level was also determined in the BG-3% group (Table 1). Aldehydes with low threshold values are formed as a result of amino acid catabolism and lipid oxidation [29]. Most aldehydes, such as pentanal, nonanal, and hexanal, are associated with certain of flavor [8]. Hexanal is considered an indicator of lipid oxidation and shows the characteristic odor of green grass [15]. The highest hexanal abundance was found in BG-3% (Table 1). Similarly, Akansel et al. [12] found that 3% BG significantly increased the TBARS value. During the production of black garlic, alcohols, ketones, and aldehydes are formed as a result of changes in lipids. Oxidative changes in lipids are also thought to be affected by these compounds [12,30,31]. In addition, although it is known that black garlic contains more antioxidants, it is stated that long-term high temperatures applied during the process can lead to losses in newly formed or released polyphenols [30]. On the other hand, it was reported that some compounds with antioxidant activity, including polyphenols, can show a prooxidant activity at high concentrations [32,33]. In fact, in this study, the use of 3% BG caused a prooxidant effect. In addition, 2-methyl-3-phenyl-propanal, not statistically significant in this study, and other aldehydes had a significant effect on HTS and were also in agreement with other studies [8,18,19,34]. Many factors, such as the type and variety of white garlic, growing conditions, and climate, affect the content and sensory properties of white garlic and, thus, significantly affect the characteristic properties of black garlic. In addition, the conditions applied in the production of black garlic (mainly temperature, relative humidity, and time) and reactions that affect the structure and nutritional components of garlic, such as the Maillard reaction, can affect the aroma profile of black garlic [3].

In this study, three ketone, four aromatic hydrocarbon, and two ester compounds were determined, and these compounds were not affected by the use of BG (Table 1) (Figure 1). Ketones are formed as a result of amino acid degradation and microbial metabolism [35]. Acetone [19] and 3-hydroxy-2-butanone were previously observed in HTS [8,18,34]. The presence of 3-hydroxy-2-butanone gives a buttery [8,35] and overripe fruit odor [35]. This compound was produced by the fermentation of carbohydrates and by the catabolism of amino acids [35]. The aromatic hydrocarbons and esters were also identified in other studies conducted on HTS [8,18,34]. Esters are produced by the esterification of alcohols and carboxylic acids [19]. In addition, they are linked to the activity of esterases in *Staphylococcus* [35].

Terpenes are the major volatile compounds in HTS, and the main sources of these compounds are spices [8,18,19]. Martin et al. [36] emphasized that the presence of terpenes is not related to ripening and microbial activity. In this study, 18 terpene compounds were identified. Of these compounds, only four compounds were statistically affected by the use of BG (Table 1). The levels of sabinene, β-pinene, and β-phellandrene decreased in the presence of BG. Eucalyptol showed the lowest level in the BG-3% group (Table 1). In this study, it is thought that the differences determined in some terpenes were due to interactions/transformations that occurred during the stages of HTS. On the other hand, total terpene abundance was not affected by BG use (Figure 1).

### 3.2. Principal Component Analysis of Volatile Compounds

Principal component analysis was carried out to evaluate the relationships between black garlic treatments and chemical groups (Figure 3). PC1 provided 75% of the variance, while PC2 provided 22%. Thus, the first two PCs provided 97% of the total variance. PC1 separated all groups containing different amounts of BG from WG-1%, and these groups were on the positive side of PC1. In contrast, WG-1% was on the negative side of PC1 and positively correlated with terpenes, aliphatic hydrocarbons, and acids. BG-containing groups showed a positive correlation with esters, sulfur compounds, aldehydes, aromatic hydrocarbons, and alcohols, but these chemical groups were more closely correlated with BG-3% (Figure 3).

The principal component analysis of the relationships between black garlic treatments and volatile compounds is given in Figure 4. PC1 and PC2 explained 66% and 27% of the variation, respectively. In other words, the first two principal components explained 93% of the total variance. WG-1% and BG-1% groups were placed on the negative side of PC1, while BG-2% and BG-3% groups were placed on the positive side of PC1. Diallyl disulfide, which is one of the main factors that causes the pungent and spicy smell of garlic, showed a close correlation with WG-1%. As a matter of fact, it was determined that this compound is more concentrated in white garlic than in black garlic [5,24,26]. Similarly, allyl mercaptan had a closer correlation with the WG-1% and BG-1% groups. Kim et al. [24] also found that allyl mercaptan is more closely related to white garlic.

## 4. Conclusions

The effect of black garlic, a functional garlic product, on volatile compounds varies depending on the type of volatile compound. The use of black garlic affected a few volatile compounds. 2-propen-1-ol increased in the presence of BG, and the use of BG reduced the level of diallyl disulfide and therefore the pungent and spicy smell of garlic. According to this result, HTS containing black garlic, which is considered a functional food, can be an alternative product for individuals affected by the pungent aroma of garlic and can be considered a novel product. However, the level of hexanal, considered an indicator of lipid oxidation, increased in the BG-3% group. For this reason, it may be considered that the use of 3% BG in HTS production is not appropriate. But studies on sensory analysis are needed to confirm this result. On the other hand, there is also a need for studies on the quality properties of HTS of the functional compounds originating from black garlic.

## Figures and Tables

**Figure 1 foods-12-03876-f001:**
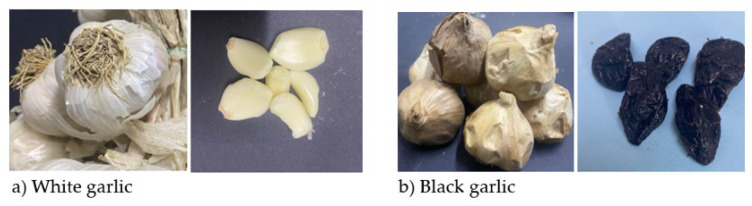
White (**a**) and black (**b**) garlic.

**Figure 2 foods-12-03876-f002:**
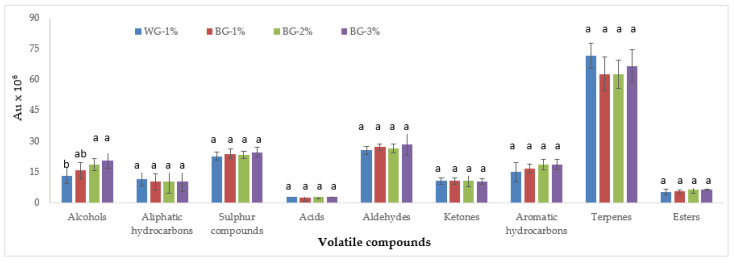
Effect of black garlic use on volatile compounds of HTS (means with the same letter within the same group are not statistically difference from each other, *p* < 0.05).

**Figure 3 foods-12-03876-f003:**
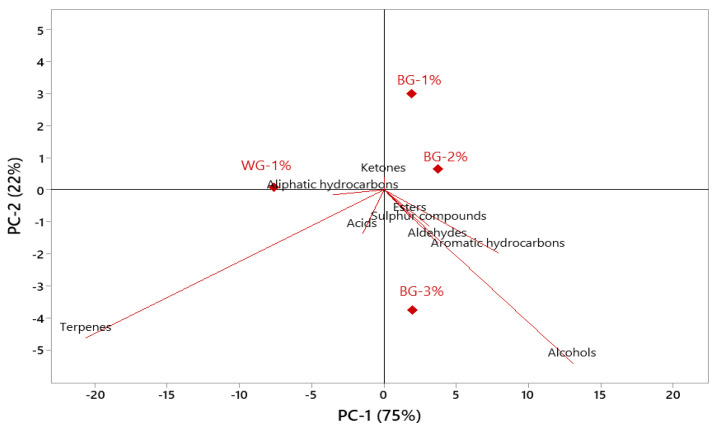
Principal component analysis of the relationships between black garlic treatments and chemical groups.

**Figure 4 foods-12-03876-f004:**
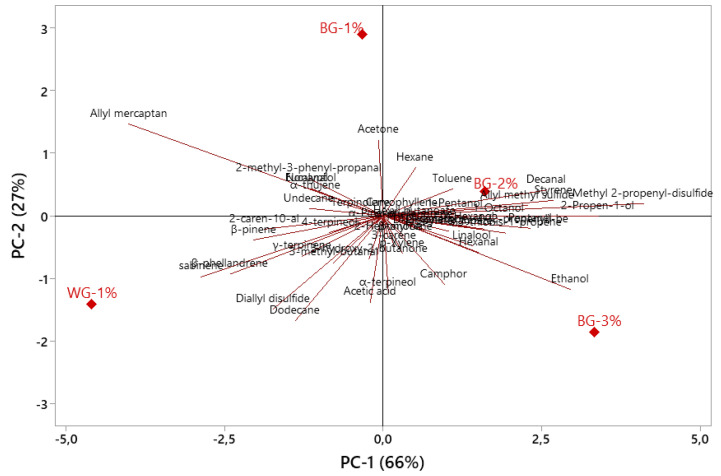
Principal component analysis of the relationships between black garlic treatments and volatile compounds.

**Table 1 foods-12-03876-t001:** Effect of using black garlic at different levels on volatile compounds of HTS (mean ± SD) (arbitrary units ×10^6^).

Compounds	RI	KI	WG-1%	BG-1%	BG-2%	BG-3%	Sig.
Alcohols							
Ethanol	a	539	3.02 ± 1.44	3.13 ± 1.75	3.90 ± 1.49	5.27 ± 1.35	ns
2-propen-1-ol	b	605	2.01 ± 0.69 b	3.28 ± 0.73 a	3.74 ± 0.62 a	4.36 ± 1.12 a	*
1-pentanol	b	835	1.99 ± 1.63	2.36 ± 1.23	2.75 ± 0.77	2.68 ± 0.81	ns
1-hexanol	a	931	2.98 ± 1.38	3.13 ± 0.81	4.02 ± 1.47	3.71 ± 1.40	ns
1-octanol	b	1132	3.22 ± 0.98	3.87 ± 1.30	4.11 ± 0.88	4.50 ± 0.54	ns
Aliphatic hydrocarbons							
Hexane	a	600	2.26 ± 2.59	3.25 ± 1.12	2.63 ± 3.67	2.64 ± 1.77	ns
Undecane	a	1100	4.14 ± 2.42	3.67 ± 1.55	3.88 ± 1.22	3.17 ± 2.19	ns
Dodecane	a	1200	5.27 ± 2.33	3.30 ± 2.62	3.71 ± 2.79	4.58 ± 3.54	ns
Sulfur compounds							
Allyl mercaptan	b	610	4.66 ± 0.63 a	4.67 ± 1.00 a	2.81 ± 1.49 b	1.93 ± 0.37 b	**
Allyl methyl sulfide	b	730	4.21 ± 0.71 b	5.22 ± 0.89 a	5.44 ± 0.84 a	5.77 ± 0.51 a	*
3,3′-thiobis-1-propene	b	888	4.01 ± 0.81	4.46 ± 1.02	4.94 ± 1.01	5.35 ± 1.11	ns
Methyl 2-propenyl-disulfide	b	946	3.59 ± 0.76 b	5.36 ± 1.05 a	5.67 ± 1.11 a	6.46 ± 0.71 a	**
Diallyl disulfide	a	1116	6.11 ± 0.89 a	4.15 ± 0.87 b	4.49 ± 1.09 b	5.13 ± 0.64 b	**
Acid							
Acetic acid	a	717	4.15 ± 0.92	2.67 ± 1.79	3.78 ± 1.71	4.00 ± 1.33	ns
Aldehydes							
3-methyl-butanal	b	693	3.59 ± 1.16	2.61 ± 0.67	2.88 ± 0.86	3.13 ± 0.47	ns
Pentanal	b	742	4.45 ± 1.01 b	5.27 ± 0.79 ab	5.29 ± 0.77 ab	6.22 ± 1.23 a	*
Decanal	a	824	2.51 ± 0.81 b	3.82 ± 0.93 a	4.07 ± 0.48 a	4.21 ± 1.16 a	*
Hexanal	a	849	4.49 ± 0.32 b	4.59 ± 0.62 b	4.87 ± 0.56 b	5.67 ± 0.91 a	*
Nonanal	a	1163	4.17 ± 1.24	4.18 ± 0.64	3.62 ± 0.79	3.36 ± 1.23	ns
2-methyl-3-phenyl-propanal	b	1334	6.50 ± 2.12	6.78 ± 2.17	5.76 ± 2.63	5.83 ± 3.76	ns
Ketones							
Acetone	a	530	3.31 ± 1.72	4.25 ± 1.91	4.05 ± 1.89	2.98 ± 1.06	ns
3-hydroxy-2-butanone	b	779	5.10 ± 0.68	4.34 ± 1.39	4.80 ± 1.07	4.97 ± 1.06	ns
2-heptanone	b	922	2.26 ± 0.66	2.04 ± 0.74	2.02 ± 0.85	2.42 ± 0.63	ns
Aromatic hydrocarbons							
Toluene	a	785	4.76 ± 0.78	5.48 ± 0.57	5.66 ± 0.96	5.38 ± 0.85	ns
P-xylene	a	892	2.48 ± 1.86	1.86 ± 1.30	2.78 ± 2.64	2.55 ± 1.11	ns
Styrene	b	916	3.93 ± 1.83	5.29 ± 0.91	5.14 ± 0.96	5.90 ± 0.86	ns
1,2-dimethoxy-4-(2-propenyl)-benzene	b	1457	3.81 ± 1.51	3.97 ± 0.85	5.05 ± 1.41	4.66 ± 1.19	ns
Terpenes							
α-thujene	b	934	3.37 ± 0.76	3.15 ± 0.49	3.19 ± 1.04	2.48 ± 1.07	ns
α-pinene	a	939	4.06 ± 0.88	3.90 ± 1.25	3.85 ± 1.05	3.98 ± 2.69	ns
Sabinene	b	954	4.56 ± 0.83 a	2.58 ± 0.44 b	2.71 ± 0.77 b	2.67 ± 1.17 b	**
β-pinene	b	987	3.07 ± 0.67 a	1.95 ± 0.63 b	1.86 ± 0.53 b	1.70 ± 0.70 b	**
β-myrcene	b	1005	3.11 ± 0.90	3.00 ± 0.62	3.01 ± 1.03	3.46 ± 1.59	ns
α-phellandrene	b	1022	2.34 ± 1.35	2.30 ± 1.02	2.74 ± 1.29	2.58 ± 1.02	ns
3-carene	b	1026	4.46 ± 1.21	4.10 ± 0.94	4.32 ± 0.98	4.65 ± 1.01	ns
D-limonene	a	1043	4.76 ± 0.67	4.82 ± 0.89	4.94 ± 0.29	5.26 ± 1.14	ns
β-phellandrene	b	1046	5.39 ± 0.76 a	3.62 ± 0.94 b	3.80 ± 0.89 b	3.81 ± 1.14 b	*
Eucalyptol	b	1054	3.32 ± 0.66 a	3.32 ± 0.25 a	2.84 ± 0.59 ab	2.52 ± 0.26 b	*
γ-terpinene	b	1071	3.26 ± 3.01	2.13 ± 0.82	2.47 ± 0.44	2.38 ± 1.04	ns
Terpinolene	b	1095	4.09 ± 0.68	4.10 ± 0.88	3.70 ± 0.63	3.96 ± 0.78	ns
Linalool	a	1142	4.91 ± 0.99	5.09 ± 0.47	5.25 ± 0.77	6.03 ± 0.57	ns
Camphor	b	1207	3.19 ± 1.89	2.44 ± 2.13	3.48 ± 3.39	3.85 ± 2.97	ns
4-terpineol	b	1220	3.52 ± 0.67	3.07 ± 1.04	2.74 ± 0.61	3.09 ± 1.95	ns
α-terpineol	b	1252	2.95 ± 2.63	1.98 ± 1.16	2.24 ± 1.22	3.27 ± 1.13	ns
2-caren-10-al	b	1336	6.69 ± 2.39	6.07 ± 3.41	5.05 ± 2.83	5.73 ± 2.94	ns
Caryophyllene	b	1490	4.75 ± 1.83	5.13 ± 2.55	4.42 ± 1.97	5.21 ± 2.05	ns
Esters							
Propyl hexanoate	a	1151	3.05 ± 0.91	3.13 ± 1.09	3.66 ± 0.81	3.70 ± 0.40	ns
Hexyl butanoate	a	1224	2.36 ± 0.89	2.45 ± 0.54	2.63 ± 0.90	2.69 ± 0.64	ns

RI: identification reliability; a: retention time identical with an authentic sample and mass spectrum; b: Kovats index from the literature in accordance and mass spectrum. KI: Kovats index was determined for DB-624 column. a,b: means with different letters in the same row are statistically different (*p* < 0.05). **: *p* < 0.01, *: *p* < 0.05, ns: not significant.

## Data Availability

Data is contained within the article.

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
