# Peer review of "The Effect of Black Garlic on the Volatile Compounds in Heat-Treated Sucuk"

_foods, 2023, doi:10.3390/foods12203876_

Round 1
Reviewer 1 Report
Introduction
Add some text about sucuk: what is difference from other sausages
Results
Need brief information about the chemical composition of control and experimental samples of sucuk.
What is the effect of adding black garlic on the degree of oxidation of fats (lipids)
Author Response
Comments and Suggestions for Authors:
Introduction
Add some text about sucuk: what is difference from other sausages
-Revised
Results
Need brief information about the chemical composition of control and experimental samples of sucuk.
-Only volatile compounds were determined.The pH value was applied to follow -up fermentation and measurement was made during fermentation. In addition, water activity measurement was followed to end the drying process. The methods of the measurement of pH and AW values are added to the text.
What is the effect of adding black garlic on the degree of oxidation of fats (lipids)
-Added
Reviewer 2 Report
Dear Editor and Authors,
the manuscript "The effect of black garlic on the volatile compounds of heat 2
treated sucuk" is certainly a valuable work that, after proofreading, will be a valuable contribution to the Foods journal.
Introduction is good written.
Materials and methods:
line 86-90: Authors wrote that they added the 20 g NaCl, 2.5 g allspice, 5 g black pepper, 7 g red pepper, 9 g cumin, 4 g sucrose, and, 0.15 g sodium nitrite - don't these additives cause the occurrence of volatile compounds? If these additives cause the presence of volatile components, the control sample should be without additives. I think in this case the sausages should be without additives - only with black garlic.
Line 90-91: The Authors wrote that "For control treatment, 1% white garlic was added to the batters. For other treatments, 1%, 2% or 3% black garlic were used.". Why does the control sample have white garlic and is compared to black garlic? I believe that the white garlic trial should not be a control trial. As I wrote above. The white garlic and black garlic were the samples and why is the control sample without this additives? I think it should be correct.
The black garlic trials are well described and the methodology is well selected. The description for black garlic is well presented. The control sample is incorrectly selected.
The work is valuable, the problem is the control sample.
Conclusion will be other in case other control sample.
Sentences are short and understandable.
Author Response
Dear Editor and Authors,
the manuscript "The effect of black garlic on the volatile compounds of heat treated sucuk" is certainly a valuable work that, after proofreading, will be a valuable contribution to the Foods journal.
Introduction is good written.
-Thank you
Materials and methods:
line 86-90: Authors wrote that they added the 20 g NaCl, 2.5 g allspice, 5 g black pepper, 7 g red pepper, 9 g cumin, 4 g sucrose, and, 0.15 g sodium nitrite - don't these additives cause the occurrence of volatile compounds? If these additives cause the presence of volatile components, the control sample should be without additives. I think in this case the sausages should be without additives - only with black garlic.
- Various spices are used in the production of semi-dry fermented sausages, such as heat-treated sucuk. Of course, these have an impact. However, the same formulation was applied in all groups, thus eliminating differences in spices. In other words, this study is not a model study. It is a production in the real system. For this reason, production without spices or garlic is not suitable for the real system. Garlic is a characteristic ingredient of this product.
Line 90-91: The Authors wrote that "For control treatment, 1% white garlic was added to the batters. For other treatments, 1%, 2% or 3% black garlic were used.". Why does the control sample have white garlic and is compared to black garlic? I believe that the white garlic trial should not be a control trial. As I wrote above. The white garlic and black garlic were the samples and why is the control sample without this additives? I think it should be correct.
The black garlic trials are well described and the methodology is well selected. The description for black garlic is well presented. The control sample is incorrectly selected.
The work is valuable, the problem is the control sample.
Conclusion will be other in case other control sample.
- White garlic is a key ingredient in heat-treated sucuk. Since production cannot be carried out without using white garlic, white garlic was chosen as the control group. Black garlic was added at 3 different levels and thus, it was aimed to determine the effect of black garlic ratio on volatile compounds. In other words, this study is not a model study. It is a production in the real system.
Reviewer 3 Report
The authors studied the influence of different levels of black garlic on the organic volatile compounds of heat-treated sucuk, a semi-dry fermented sausage from Turkey.
The study is interesting. But it should be improved. My comments are as follows:
English may be improved.
Reference numbers in the text: the authors must follow the “Instructions for Authors”.
In the “Introduction” the authors state that “strategies are being developed to reduce synthetic additives or the use of natural products with functional components in order to increase the value-added of these products”.
In the "Discussion" there is no mention of this aspect. Why not?
In lines 49-50 the authors report that “….this situation increases the demand for alternative garlic products such as black garlic, which do not have an intense odor”. This aspect is not taken into consideration at all.
Line 53: It would be better if the authors briefly described sucuk, and heat-treated sucuk.
The authors define black garlic as a functional garlic product. In lines 56-57 the authors state that “….On the other hand, the use of black garlic in these products is also on the agenda due to its functional properties and popularity”. There is no comment on this aspect.
Lines 67-69: The objective of the study is not clear enough. The authors have emphasized the functional properties of the black garlic and the effect of the black garlic on the formation of nitrosamines in the HTS and on some of the product properties. My question is: "Why do the authors only report nitrosamine formation? Akansel et al.2023 [12] studied the” Effect of black garlic on microbiological properties, lipid oxidation, residual nitrite, nitrosamine formation and sensory characteristics in a semi-dry fermented sausage”.
Line 89: “Starter cultures”. My questions are: “Were the starter cultures lyophilized or added in some other form? What was the bacterial load?”. It would be better to have an indication of the form in which the starter cultures have been used and the bacterial load.
Line 96: “pH reached 5.3.”. It would be better to specify the pH measurement method.
Line 151: “[…18-20]. The references numbers must be“[…18-19].
Line 152: the authors do not consider "1-octanol". Its concentration is higher than that of "2-propen-1-ol" in BG-1%, BG-2%, and BG-3%. This compound may be linked to Latiplantibacillus plantarum metabolism.
My question is: what is the reason why the authors did not consider a correlation between starter cultures and volatile compounds? The metabolic activity of starter cultures can be of great importance in the formation of volatile compounds.
Further comments on the origin of the volatile compounds of the samples should be added to the Considerations.
Moreover, many volatile compounds give specific aromatic "notes" to food. The authors have not considered this aspect. Their comments seem to be incomplete, and they need to be improved.
Conclusions need to be improved according to the results obtained.
Another question I have is: what are the practical implications of this study? It would be better to specify this (e.g. influence on consumer choice; practical benefits for sucuk producers, etc.).
English may be improved
Author Response
Reviewer 3 comments:
The authors studied the influence of different levels of black garlic on the organic volatile compounds of heat-treated sucuk, a semi-dry fermented sausage from Turkey.
The study is interesting. But it should be improved. My comments are as follows:
English may be improved.
-Checked
Reference numbers in the text: the authors must follow the “Instructions for Authors”.
-Checked
In the “Introduction” the authors state that “strategies are being developed to reduce synthetic additives or the use of natural products with functional components in order to increase the value-added of these products”.
In the "Discussion" there is no mention of this aspect. Why not?
-Since there were no analysis regarding functional features in the study, no explanation was given in the discussion section.
In lines 49-50 the authors report that “….this situation increases the demand for alternative garlic products such as black garlic, which do not have an intense odor”. This aspect is not taken into consideration at all.
- The odor of black garlic is not very intense. Therefore, in this study, the black garlic content was increased up to 3%.
Line 53: It would be better if the authors briefly described sucuk, and heat-treated sucuk.
- Added.
The authors define black garlic as a functional garlic product. In lines 56-57 the authors state that “….On the other hand, the use of black garlic in these products is also on the agenda due to its functional properties and popularity”. There is no comment on this aspect.
-The sentence was removed.
Lines 67-69: The objective of the study is not clear enough. The authors have emphasized the functional properties of the black garlic and the effect of the black garlic on the formation of nitrosamines in the HTS and on some of the product properties. My question is: "Why do the authors only report nitrosamine formation? Akansel et al.2023 [12] studied the” Effect of black garlic on microbiological properties, lipid oxidation, residual nitrite, nitrosamine formation and sensory characteristics in a semi-dry fermented sausage”.
- The objective of the study was revised.
In addition, for the study of Akansel et al. (2023), not only nitrosamins but also other product characteristics were mentioned. This statement “some product properties” was used.
Line 89: “Starter cultures”. My questions are: “Were the starter cultures lyophilized or added in some other form? What was the bacterial load?”. It would be better to have an indication of the form in which the starter cultures have been used and the bacterial load.
-Revised
Line 96: “pH reached 5.3.”. It would be better to specify the pH measurement method.
-Added
Line 151: “[…18-20]. The references numbers must be“[…18-19].
-The reference numbers were controlled. 18,19 and 20th references were showed as 18-20.
Line 152: the authors do not consider "1-octanol". Its concentration is higher than that of "2-propen-1-ol" in BG-1%, BG-2%, and BG-3%. This compound may be linked to Latiplantibacillus plantarum metabolism.
My question is: what is the reason why the authors did not consider a correlation between starter cultures and volatile compounds? The metabolic activity of starter cultures can be of great importance in the formation of volatile compounds.
-In the study, same starter cultures were used in all groups.
Therefore, it is not possible to associate the results with starter cultures. In addition, 1-octanol was not found to be statistically significant. Of the alcohols, only 2-propen-1-ol was significant (Table 1).
Further comments on the origin of the volatile compounds of the samples should be added to the Considerations.
Moreover, many volatile compounds give specific aromatic "notes" to food. The authors have not considered this aspect. Their comments seem to be incomplete, and they need to be improved.
-Added
Conclusions need to be improved according to the results obtained.
Another question I have is: what are the practical implications of this study? It would be better to specify this (e.g. influence on consumer choice; practical benefits for sucuk producers, etc.).
-Revised.
Round 2
Reviewer 2 Report
Thank You for response. Now it is ok for me.
it is ok
Author Response
Reviewer 2
Comments and Suggestions for Authors: Thank You for response. Now it is ok for me.
-Thank you.
Comments on the Quality of English Language: it is ok
-Thank you.